# Compositional Lotka-Volterra describes microbial dynamics in the simplex

**Tyler A. Joseph**[1], **Liat Shenhav**[2], **Joao B. Xavier**[3], **Eran Halperin**[2,4,5,6,7], **Itsik Pe'er**[1,8,9]*

**1** Department of Computer Science, Columbia University, New York, New York, United States of America, **2** Department of Computer Science, UCLA, Los Angeles, California, United States of America, **3** Memorial Sloan Kettering Cancer Center, New York, New York, United States of America, **4** Department of Human Genetics, University of California Los Angeles, Los Angeles, California, United States of America, **5** Department of Anesthesiology and Perioperative Medicine, University of California Los Angeles, Los Angeles, California, United States of America, **6** Department of Computational Medicine, UCLA, Los Angeles, California, United States of America, **7** Institute of Precision Health, UCLA, Los Angeles, California, United States of America, **8** Department of Systems Biology, Columbia University, New York, New York, United States of America, **9** Data Science Institute, Columbia University, New York, New York, United States of America

* itsik@cs.columbia.edu

**Data Availability Statement:** All data are public; code to replicate experiments is available at https://github.com/tyjo/clv.

**Funding:** TAJ was supported by NSF fellowship DGE-1644869. IP was supported by NSF CCF-

## Abstract

Dynamic changes in microbial communities play an important role in human health and disease. Specifically, deciphering how microbial species in a community interact with each other and their environment can elucidate mechanisms of disease, a problem typically investigated using tools from community ecology. Yet, such methods require measurements of absolute densities, whereas typical datasets only provide estimates of relative abundances. Here, we systematically investigate models of microbial dynamics in the simplex of relative abundances. We derive a new nonlinear dynamical system for microbial dynamics, termed "compositional" Lotka-Volterra (cLV), unifying approaches using generalized Lotka-Volterra (gLV) equations from community ecology and compositional data analysis. On three real datasets, we demonstrate that cLV recapitulates interactions between relative abundances implied by gLV. Moreover, we show that cLV is as accurate as gLV in forecasting microbial trajectories in terms of relative abundances. We further compare cLV to two other models of relative abundance dynamics motivated by common assumptions in the literature—a linear model in a log-ratio transformed space, and a linear model in the space of relative abundances—and provide evidence that cLV more accurately describes community trajectories over time. Finally, we investigate when information about direct effects can be recovered from relative data that naively provide information about only indirect effects. Our results suggest that strong effects may be recoverable from relative data, but more subtle effects are challenging to identify.

## Author summary

Dynamic changes in microbial communities play an important role in human health and disease. Specifically, deciphering how microbial species in a community interact with

1547120, NSF DGE-1144854, and NIH U54CA209997. EH and LS were partially supported by the National Science Foundation (grant no. 1705197) and by NIH 1R56MD013312. EH was also partially supported by NIH/NHGRI HG010505-02, NIH 1R01MH115979, NIH 5R25GM112625, and NIH 5UL1TR001881. The funders had no role in study design, data collection and analysis, decision to publish, or preparation of the manuscript.

**Competing interests:** The authors have declared that no competing interests exist.

each other and their environment can elucidate mechanisms of disease, a problem typically investigated using tools from community ecology. Yet, such methods require measurements of absolute densities, whereas typical only provide estimates of relative abundances. We investigate methods for describing microbial dynamics in terms of relative abundances using approaches from machine learning and dynamical systems. Across three real datasets, we show that relative abundances are sufficient to describe compositional dynamics. Additionally, we show that models trained on relative abundances alone predict future compositions as well models trained on absolute abundances. Finally, we provide criteria for when direct effects, which typically can only be learned from absolute abundances, are recoverable for relative data. As a proof of concept, we recapitulate a previously proposed interaction network for *C. difficile* colonization.

## Introduction

The completion of the second phase of the Human Microbiome Project has highlighted the relationship between dynamic changes in the microbiome and disease [1]. Temporal changes in the vaginal microbiome during pregnancy, for example, are associated with increased risk for preterm birth [2], and the intestinal microbiome of individuals with inflammatory bowel disease undergoes large-scale changes during active and inactive periods of the disease [3]. Changes to the intestinal microbiome are also predictive of treatment outcomes. In hospitalized patients the intestinal microbiome provides resistance to pathogenic bacteria, and depletion of the community in response to antibiotics increases risk of infection [4, 5]. Moreover, some treatments for disease are mediated through the intestinal microbiome itself [6]. Consequently, recent research has focused on temporal modeling of the microbiome with the aim of understanding the etiology of disease, predicting patient outcomes for personalized medicine, and elucidation of microbe-microbe or host-microbiome interactions [5, 7, 8]. Yet, the gut microbiome is a complex ecosystem, making realization of these goals a challenging task.

Approaches to modeling microbial dynamics originate across different domains with different goals (e.g. community ecology, applied statistics, and compositional data analysis), and the field has yet to converge on a standard methodology for time-series modeling. For instance, traditional approaches based in community ecology describe temporal dynamics using generalized Lotka-Volterra (gLV) differential equations (e.g. [7–10]): a system of nonlinear differential equations modeling competitive and cooperative interactions, extended to include external perturbations by Stein *et al.* [7]. Such models have been shown to accurately predict community dynamics [7, 10]. However, gLV-based models describe dynamics in terms of absolute densities of taxa. Estimating model parameters requires measurements of community size—either from quantitative PCR, spiked-in samples of known concentrations, fluorescence-activated cell sorting, or other cell counting methods—in addition to sequencing counts of constituent taxa [11]. While measurements of community size are often required to infer direct interactions and effects, they are often unavailable.

In parallel, there has been an increasing appreciation for the compositional nature of many microbial datasets [12–15], motivating research that explores the boundaries of inference from sequencing counts alone (e.g. [16–18]). Sequencing counts only contain noisy information about the relative abundances of community members: the total number of sequencing reads is independent of the size of the community. Approaches to time-series modeling from relative abundances generally fall in to two categories: linear models using relative abundances, and linear models using a compositional data transformation. For the former, McGeachie *et al.* [19]

and Gibbons *et al.* [20] model relative abundance trajectories as linear combinations of previous time points and external effects. However, these methods implicitly ignore the constraint that relative abundances must sum to one and are therefore negatively correlated, making parameter estimates difficult to interpret. Li et al. [34] suggest addressing this by inference of the latent overall biomass. Alternatively, Shenhav *et al.* [21] suggested a linear mixed model with variance components, while representing the previous state microbial community using its quantiles instead of relative abundances. Yet, binning taxa into quantiles may lose fine-grained information about interactions. Indeed, correctly modeling relative abundance data is challenging because the data is in a constrained space (the simplex, where relative abundances must sum to one), which can lead to spurious associations if standard statistical tools are applied directly [14].

A promising alternative uses methods from compositional data analysis [22, 23], a branch of statistics devoted to the analysis of simplex-valued data. Techniques from compositional data analysis alleviate problems of working in the simplex by transforming data from a constrained space to an unconstrained one with orthogonal coordinates and statistically independent components. Specifically, relative abundances are transformed to log-ratios using a compositional data transformation, such as the isometric log-ratio transformation [24] or additive log-ratio transformation [22]. For example, Silverman *et al.* [25] combine a phylogenetically motivated log-ratio transformation (PhyILR, [18]) with dynamic linear models to describe microbial dynamics. Äijö *et al.* [26] similarly provide a correction for sequencing noise by using a Gaussian process model to parameterize a multinomial distribution on sequencing counts, thereby providing a statistical correction for zero-inflation and over-dispersion common to microbial datasets. Their model implicitly describes dynamics in an additive log-ratio transformed space. Alternatively, Jarauta and Egozcue [27] investigate predator-prey interactions using simplical linear differential equations coupled with separate equations for community size. Their approach, however, is restricted to oscillatory behavior of competitive interactions between two-species. Nonetheless, much remains unknown about the compositional dynamics of the microbiome, and few studies compare these diverse approaches on equal footing.

In this paper, we investigate models of microbial dynamics in the simplex. As a guiding principle, we derive a new dynamical system for simplex-valued data from the generalized Lotka-Volterra equations, which we term "compositional" Lotka-Volterra (cLV), synthesizing approaches from community ecology and compositional data analysis. On three real datasets, we show that the parameters of cLV recapitulate interactions in the simplex implied by gLV. Moreover, we show that cLV is as accurate as gLV in forecasting microbial trajectories in terms of relative abundances, suggesting that estimated concentrations are unnecessary for predicting community trajectories in terms of relative abundances. We further compare cLV to two other models of relative abundance dynamics: a linear models under the additive log-ratio transformation, and a linear on relative abundances. We provide evidence that cLV better describes community dynamics than linear models, suggesting that nonlinear models are important for accurately describing community dynamics in the simplex. Finally, we investigate when direct effects can be recovered from relative data. We provide a proof-of-concept demonstration where we recapitulate a proposed interaction network with *C. difficile* inferred using absolute densities.

## Results

### Motivation

Our motivation for this work is three-fold. First, gLV has a strong theoretical foundation in community ecology and dynamical systems theory, and has been shown to accurately describe

community dynamics of the microbiome. However, gLV models absolute abundances, and we would like to extend the model to relative abundances. Second, approaches from compositional data analysis have highlighted the benefits—both statistical and practical—of transforming constrained relative abundances to an unconstrained space using a log-ratio transformation. Thus, we would like to express microbial dynamics under such a transformation. Third, several models for the dynamics of relative abundances exist in the literature, most of which are linear models using relative abundances or linear models in a transformed space. Yet if we believe gLV, a nonlinear model, then linear models will fail to accurately describe community dynamics and predict community changes. Hence, we want to compare linear and nonlinear models. We emphasize that directly applying gLV to relative abundances lacks mathematical justification. Specifically, gLV models the change in the absolute abundance of each taxon over time, whereas the appropriate model for relative abundances derived from gLV results in equations that depend on total community size. This means an approximation is required. In the following sections we develop such an approximate model, devise a method to infer its parameters, and explain its correspondence with gLV.

## Compositional Lotka-Volterra

The gLV equations describe the dynamics of microbial taxa in terms of their concentration or density, i.e., number of cells per unit of volume. We standardly denote by $x_i(t)$ the concentration of taxon $i$ at time $t$ for the chosen scale for $i = 1, \ldots, D$, and let $u_p(t)$ be an indicator variable describing presence or absence of external perturbation $p$ at time $t$ for $p = 1, \ldots, P$. These equations thus state that the change in concentration of $x_i(t)$ is determined by a taxon specific growth rate $g_i$, interactions between taxa $A_{ij}$, and the effect of each external perturbation $p$ on each taxon $i$, $B_{ip}$. Specifically, gLV-based models use the following set of nonlinear (Riccati) differential equations:

$$\forall i = 1, \ldots, D : \frac{d}{dt} x_i(t) = x_i(t) \left( g_i + \sum_{j=1}^{D} A_{ij} x_j(t) + \sum_{p=1}^{P} B_{ip} u_p(t) \right). \tag{1}$$

Assuming $x_i(t) > 0$, equivalent equations describe compositional dynamics under the additive log-ratio transformation:

$$\forall i = 1, \ldots, D : \frac{d}{dt} \log x_i(t) = g_i + \sum_{j=1}^{D} A_{ij} x_j(t) + \sum_{p=1}^{P} B_{ip} u_p(t) \tag{2}$$

Define $N(t) = \sum_{j=1}^{D} x_j(t)$ and $\pi_i(t) = \frac{x_i(t)}{N(t)}$. Then, using the additive log ratio (alr) transformation

$$\frac{d}{dt} \log\left( \frac{\pi_i(t)}{\pi_D(t)} \right) = \overbrace{(g_i - g_D)}^{\bar{g}_i} + \sum_{j=1}^{D} \overbrace{(A_{ij} - A_{Dj})}^{\bar{A}_{ij}} x_j(t) + \sum_{p=1}^{P} \overbrace{(B_{ip} - B_{Dp})}^{\bar{B}_{ip}} u_p(t) \tag{3}$$

$$= \bar{g}_i + \sum_{j=1}^{D} N(t) \bar{A}_{ij} \pi_j(t) + \sum_{p=1}^{P} \bar{B}_{ip} u_p(t) \tag{4}$$

The volume scale of $x_i(t)$ is arbitrary (it is defined when a measurement is taken and can be rescaled), so without loss of generality we pick a scale such that the mean community size is 1

(i.e. $\mathbb{E}[N(t)] = 1$). Hence

$$\frac{d}{dt} \log\left(\frac{\pi_i(t)}{\pi_D(t)}\right) \approx \bar{g}_i + \sum_{j=1}^{D} \bar{A}_{ij}\pi_j(t) + \sum_{p=1}^{P} \bar{B}_{ip}u_p(t) =: f_i(t) \tag{5}$$

The terms $\bar{g}_i$, $\bar{A}_{ij}$, and $\bar{B}_{il}$ now describe relative (to the denominator) growth rates, relative interactions, and relative external effects respectively.

The additive log-ratio transformation makes explicit that model parameters describe changes ratios of taxa, the only information provided by relative abundances. Growth rates, interactions, and external perturbations can all be reasoned about through their effect on the log ratio between pairs of taxa. While the choice of denominator in the additive log-ratio transform was arbitrary, knowledge of the parameters for one choice of denominator provides information about how the ratios of all pairs of taxa change (see S1A Appendix). This means that if we are interested in the ratio of two particular taxa, we only need to learn model parameters once, then transform the system to the appropriate parameters.

We refer to Eq 5 as "compositional" Lotka-Volterra (cLV). Notably, solving for $\frac{d}{dt}\pi_i$ (see S1B Appendix) gives

$$\forall i = 1, \ldots, D-1 \; : \; \frac{d}{dt}\pi_i(t) = \pi_i(t)\big(f_i(t) - \bar{f}(t)\big) \tag{6}$$

where

$$\bar{f}(t) := \sum_{k=1}^{D-1} \pi_k(t)f_k(t) \tag{7}$$

The first set of terms of Eq 6, $\pi_i(t)f_i(t)$, correspond to gLV on relative abundances, while the second set of terms $-\pi_i(t)\bar{f}(t)$ serve as a "compositional correction:" a correction to the dynamics of $\pi_i(t)$ due to constraint that the $\pi_i(t)$ must sum to one. Fig 1 depicts examples of the phase space for compositional Lotka-Volterra.

There are several remarks to make about this derivation in light of our motivation:

- cLV is an approximation to gLV when the variance in community size, $\mathrm{Var}(N(t)) = \mathbb{E}[(N(t)-1)^2]$, is low. Then, the parameters of cLV approximately correspond to differences in parameters of gLV. For example, the interaction term $\bar{A}_{ij} = A_{ij} - A_{Dj}$ is the absolute interaction between taxon $i$ and taxon $j$ minus the effect of the denominator $D$ on taxon $j$. We suggest that a useful metric for determining when the parameters correspond is a type of "signal-to-noise" ratio for community size. Specifically,

$$\mathrm{SNR} = \frac{\mathbb{E}[N(t)]}{\sqrt{\mathrm{Var}(N(t))}} = \frac{1}{\sqrt{\mathrm{Var}(N(t))}}$$

Thus, "noise" will dominate the "signal" when $Var(N(t)) > 1$ and parameter estimates will diverge. We provide empirical evidence for this claim in the section **Correspondence with the parameters of gLV**.

- While cLV is an approximation to gLV, the two models are distinct. Mathematically, Eq 5 defines its own stand-alone dynamical system. It is therefore interesting to investigate which of these models more accurately describes relative abundance dynamics, and whether other models of relative abundance dynamics are potentially better. We focus on this question in the section **Model comparison**.

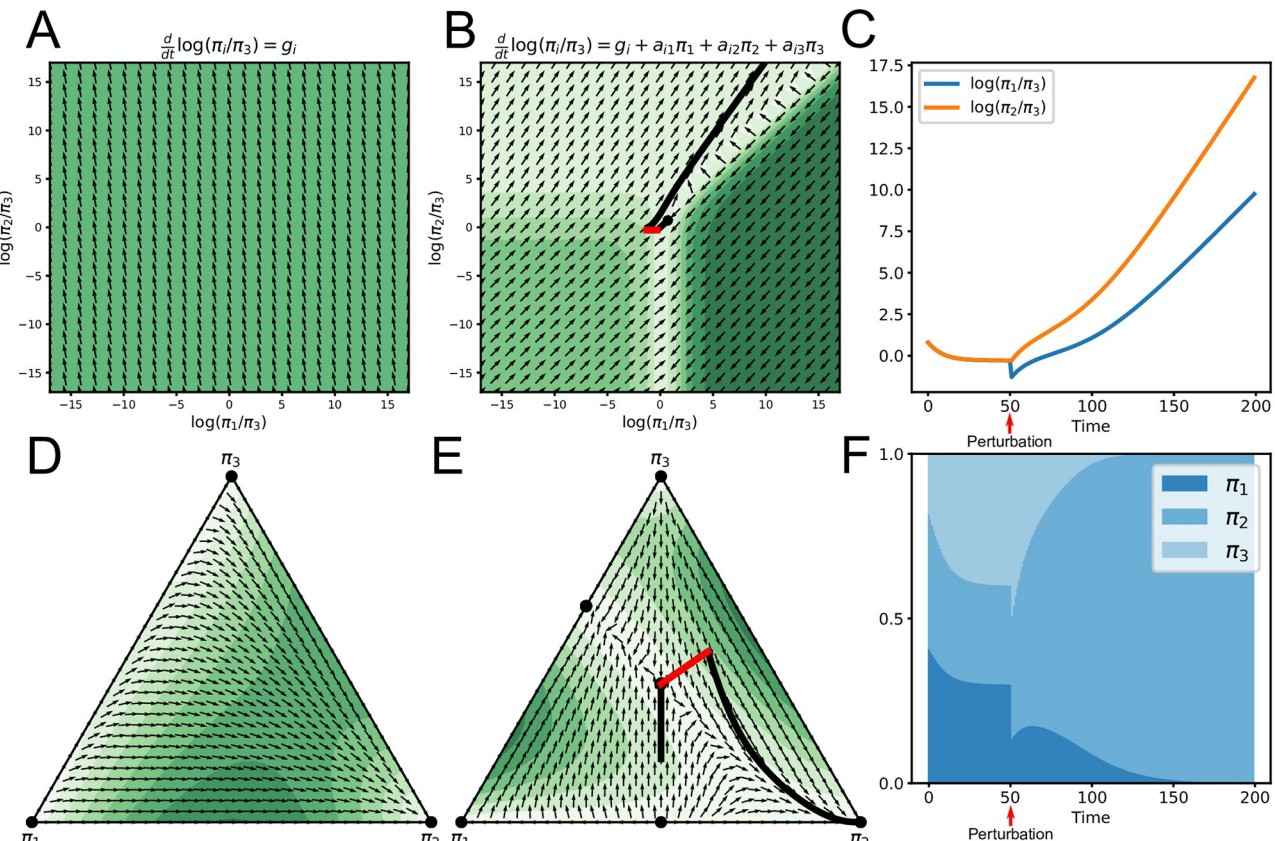

**Fig 1. Example phase spaces and trajectories for compositional Lotka-Volterra.** The top row displays examples using the additive log-ratio transformation for three taxa. The bottom row displays examples in relative-abundance space corresponding to the examples in the top row. Arrows display the direction of the gradient, while the colors display its magnitude (lighter is smaller). A) Phase space where dynamics depend on relative growth rates alone. B) Phase space where dynamics depend on growth rates and interactions. The black dot denotes a fixed point. The solid line depicts a simulated trajectory, where an external perturbation (red line) moves the system away from the fixed point. C) Alternate view of the example in B. The red arrow denotes an external perturbation causing the system to move away from the fixed point. D) Phase space in the simplex corresponding to A. Black dots denote fixed points and only occur on the corners. E) Phase space in the simplex corresponding to B. Black dots denote six possible fixed points. The line displays the same simulated trajectory as B. F) Simulation example corresponding to B depicted as relative abundances. An external perturbation at time point 50 causes the system to move away from the fixed point in the interior of the simplex toward one at the boundary.

- The correspondence between the parameters of cLV and gLV gives mathematical criteria for which absolute terms are recoverable from relative data. For interactions specifically, if the magnitude of the interaction $A_{ij}$ is larger than the magnitude of the interaction $A_{Dj}$, then cLV will recapitulate the sign of the absolute interaction. This also suggests that the appropriate choice of denominator for the alr is one where the taxon $\pi_D$ is approximately log constant over time. We investigate this in the section **Interpreting model parameters**.

We note that our derivation using the alr transformation is easily extended to other compositional data transformations, since the alr can be expressed as a linear transformation of the centered log-ratio transformation and of the isometric log-ratio transformation [24]. We derive the this transformation in S1C Appendix. This opens the door for other domain-specific data transformations, such as the phylogenetic isometric log-ratio transformation [18]. Additionally, taking the alr (or other) transformation has the benefit of expressing the system as an affine function of relative abundances, making it amenable to fast parameter inference procedures using least squares.

## Parameter inference

To examine each remark of our derivation we need to infer model parameters on real data. Thus, we first investigated methods for parameter inference. The challenge in inferring model parameters is that the number of parameters often greatly outnumbers the sample size. Therefore some form of regularization is required to avoid overfitting. Previous work on inferring parameters of gLV used ridge regression [7] for regularization, and pseudo-counts to address sequencing noise and zeros. Later work by Bucci *et al.* [10] included more sophisticated methodologies for modeling sequencing noise, however the form of our model (Eq 6) does not allow us to readily apply these methods. We therefore chose to focus on two standard approaches to regularization: ridge regression [28] and elastic net [29].

We evaluated each method by simulating data under cLV, varying sample size, sequencing depth, and frequency of longitudinal samples (see Methods). Performance was measured using three metrics: root-mean-square-error (RMSE) between estimated and ground truth interactions, RMSE between estimated and ground truth growth rates, and prediction of unseen held out trajectories from initial conditions. As a baseline, we evaluated both models using simulations with and without sequencing noise. Importantly, the choice of denominator for our simulations was arbitrary.

In our simulations without sequencing noise, elastic net regularization outperformed ridge regression, particularly at low sample sizes (Fig 2). With the introduction of sequencing noise the performance difference between models was negligible (S1 Fig). Since elastic net regularization outperformed ridge regression on the simulations without sequencing noise, and because elastic net regularization includes ridge regression as a special case, we choose to focus on elastic net regularization for the remaining simulations.

Simulations investigating sequencing depth demonstrated no noticeable gain in estimation accuracy beyond a depth of 10000 sequencing reads (S2 Fig). Nonetheless, accuracy was poorer than the simulations without sequencing noise. Simulations investigating temporal density revealed that sample size was more important than density in time. At smaller sample sizes (5 and 10 samples), predictive performance on hold out trajectories started decline when the time between samples was approximately 4 days apart (S3 Fig). At larger sample sizes, we observed little difference in ability to estimate model parameters or predictive accuracy.

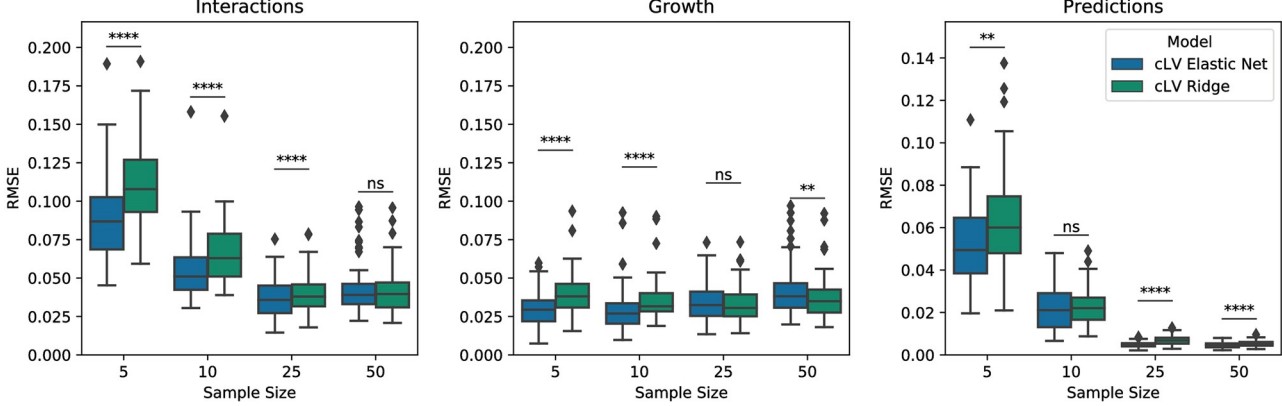

**Fig 2. Comparison of performance between ridge regression and elastic net.** Performance was evaluated on simulated ground truth using three metrics: root-mean-square error (RMSE) between true and estimated interactions, RMSE between true and estimated growth rates, and RMSE between true and estimated hold out trajectories for 5 samples per simulation replicate. Box plots describe the distribution in RMSE over 50 simulation replications. Significance is computed using the Wilcoxon signed-rank test (****: $p < 10^{-4}$; ***: $p < 10^{-3}$; **: $p < 10^{-2}$; *: $p < 0.05$; ns: not significant).

We also performed simulations to evaluate how choice of denominator affected parameter inference. In particular, we wanted to ensure that different choices of denominator do not affect quality of inference. To do this, we simulated data under one choice of denominator, performed inference with each taxon in the denominator, and computed the RMSE between inferred parameters and simulation parameters transformed to the appropriate denominator (see S1A Appendix). Quality of inference was assessed by computing the variance in RMSE of parameter estimates. A low variance suggests that inference is robust to choice of denominator. S4 Fig displays box plots of the variance in RMSE across 20 simulation replicates. Notably, in all cases the variance in estimates was low, particularly when assessing predictive performance. We note that in our simulations we enforced constraints that each taxon existed at each time point, and that its mean abundance across observations was greater than 0.001. Thus, our results suggest so long as these criteria hold parameter estimates and prediction ability are robust to choice of denominator.

## Correspondence with the parameters of gLV

We remarked that the parameters of cLV approximate differences between parameters of gLV, and that the strength of this approximation depends on variability of community size. Therefore, we next empirically investigated how well the approximation holds on three real datasets. Close correspondence between parameters suggests that relative interactions learned by cLV are representative of the true underlying relative interactions between taxa (if we treat gLV as the ground truth). Moreover, if this correspondence holds, it is suggestive of when absolute interactions are recoverable from relative data, which we examine in the section **Interpreting model parameters**.

We used three real datasets of mouse models (Table 1) that measured community density of the gut microbiome, giving estimates of both relative abundances and bacterial concentrations. The "Antibiotic" dataset consisted of 3 populations of mice (9 mice total) used to study susceptibility to *C. difficle* infection following administration of antibiotics. The "Diet" dataset included 7 mice: 5 mice were fed from a high-fiber diet, switched to a low-fiber diet, then returned to the high-fiber diet. The remaining mice were fed the high-fiber diet. In the "*C. diff*" dataset, 5 gnobiotic mice were orally gavaged with a bacterial mixture and subjected to a *C. difficle* challenge after 28 days. Administration of antibiotics, change in diet, and introduction of *C. difficle* were treated as external perturbations (in addition to including *C. difficle* as a taxon in the *C. diff* dataset). The *C. diff* dataset and Antibiotic dataset combined 16S sequencing with qPCR to estimate relative abundances and community size separately, while the Diet dataset used qPCR for individual taxa to measure concentrations. Thus, the Antibiotic, Diet, and *C. diff* datasets display a wide range of dynamics, from strong community shifts due to antibiotics, to relatively stable dynamics.

**Table 1. Description of the three real datasets investigated.**

| Name | Sample Size | Description |
|---|---|---|
| Antibiotic | 9 mice (77 total obs) | Antibiotic treated mice [7, 30]. Data collected using 16S rDNA sequencing and qPCR for biomass quantification. |
| Diet | 7 mice (330 total obs) | High-fiber to low-fiber diet and back [10]. Data collected using qPCR with taxon specific primers. |
| *C. diff* | 5 mice (130 total obs) | *C. difficle* challenge after 28 days [10]. Data collected using 16S rDNA sequencing and qPCR for biomass quantification. |

Sample sizes are listed along with the total number of observations across samples.

We trained cLV on relative abundances, and gLV on rescaled concentrations. We rescaled observed concentrations such that the average community size, $N(t)$, was 1 across observed samples. We note that this only rescales parameter estimates, and results in no loss of generality. Variance in $N(t)$ across samples is an estimator for variance in community size. For this particular task, we chose to use ridge regression since elastic net may choose to zero out different parameters for each model, making direct comparison challenging.

We observed a strong correspondence between the relative parameters estimated using cLV and parameters from gLV (Fig 3). As expected, the strength of the correlation between parameter estimates corresponded with the observed variability in community size ($\mathrm{Var}(N(t))$). The strongest correlation was observed on the Diet dataset (Pearson $r = 0.96$ for interactions, $r = 0.993$ for external effects, $r = 0.909$ for growth rates), where the size of the community remained stable (estimated $Var(N(t)) = 0.055$), while the weakest correlation was observed on the Antibiotic dataset (Pearson $r = 0.808$ for interactions, $r = 0.995$ for external effects, $r = -0.141$ for growh rates), where community size fluctuated rapidly after antibiotic administration (estimated $Var(N(t)) = 1.10$). Nonetheless, correspondence between interactions and external effects was strong among the three datasets we explored.

## Model comparison

cLV and gLV describe dynamics over time with respect to difference spaces. cLV describes relative abundances, while gLV describes absolute abundances. Additionally, there have been several other models of relative abundance dynamics proposed in the literature. A natural question is therefore: which model better describes trajectories of *relative* abundances? We thus compared cLV to gLV, and to two other models similar to others proposed in the literature:

$$\text{ALR} \qquad \frac{d}{dt}\eta_i(t) = \bar{g}_i + \sum_{j=1}^{D-1} \bar{A}_{ij}\eta_j(t) + \sum_{p=1}^{P} \bar{B}_{ip}u(t) \qquad (8)$$

$$\text{linear} \qquad \frac{d}{dt}\pi_i(t) = g_i + \sum_{j=1}^{D} A_{ij}\pi_j(t) + \sum_{p=1}^{P} B_{ip}u(t) \qquad (9)$$

where $\eta_i(t) = \log(\pi_i(t)/\pi_D(t))$, the additive log-ratio transformation. The first model (ALR) is a linear model under the additive log-ratio transformation. The second model (linear) is a linear model in relative abundances. We further compared cLV to gLV in two ways: inferring parameters on estimated concentrations (gLV$_{abs}$), and inferring parameters on estimated relative abundances (gLV$_{rel}$). The latter is equivalent to assuming constant community size. Table 2 displays the number of parameters of each model.

We evaluated models based on a measure of generalization performance, or the ability to predict unseen data. Generalization performance metrics inherently penalize models that overfit, or use parameters to fit noise in data rather than model actual signal. These metrics allow for principled comparison of models with different structures or numbers of parameters. In our case, we used a metric that evaluated predicted trajectories on held out test data via leave-one-out cross validation. That is, for each dataset we held out one time-series in turn, and trained models on the remaining data.

We further fit each model using elastic net to avoid differences in performance due to different inference procedures. Specifically, we wanted to avoid a scenario where one model outperformed the others because it used a better inference procedure. Performance was evaluated by computing the RMSE between the held-out ground truth and predicted trajectories. As a

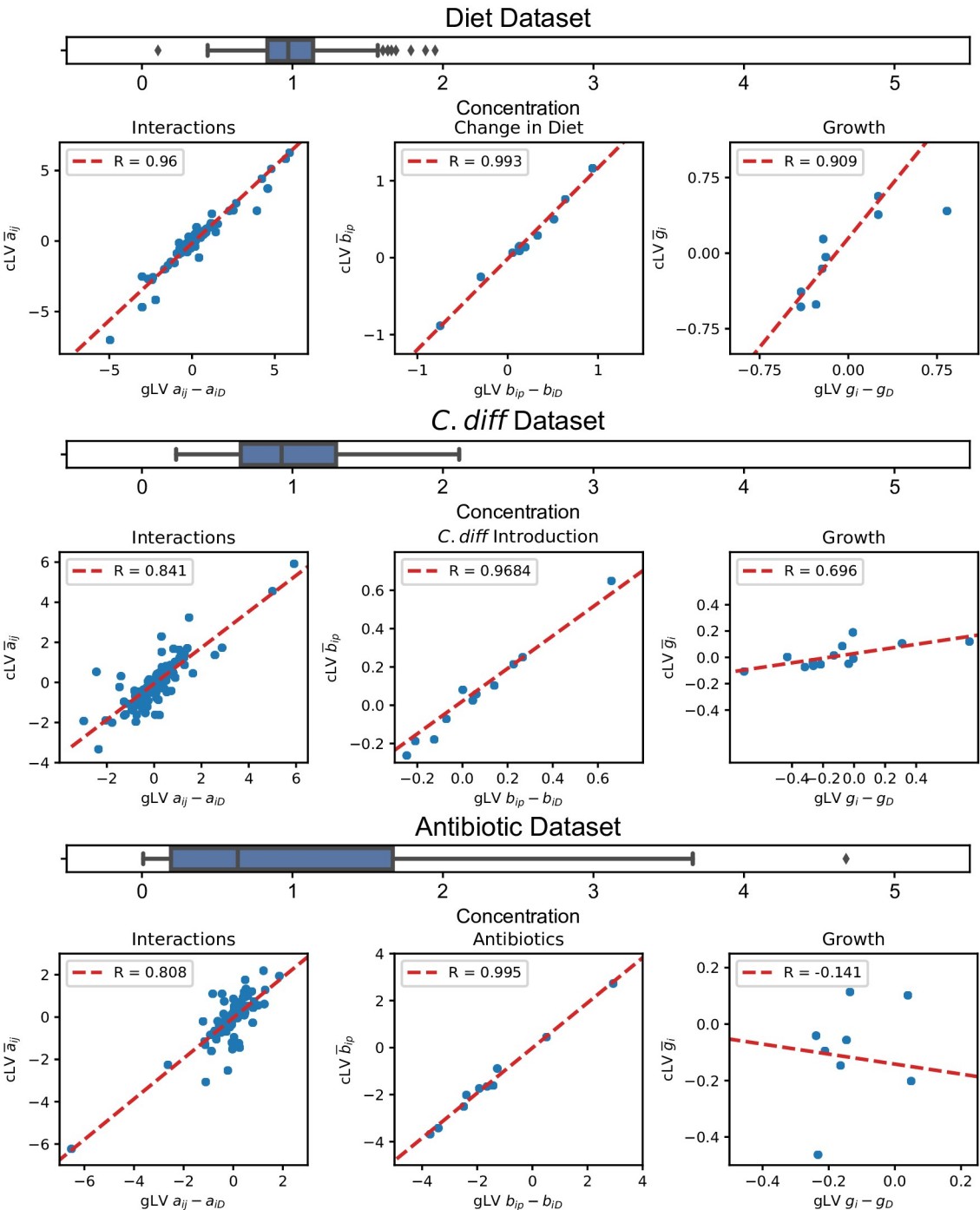

**Fig 3. Correspondence between relative parameters estimated using gLV and cLV on three datasets.** Each box plot displays the distribution of observed community size (i.e. $N(t)$) across all samples, rescaled such that $\mathbb{E}[N(t)] = 1$. Scatter plots display the relative parameters estimated by cLV ($y$-axis), and the corresponding difference in parameters of gLV ($x$-axis). cLV better approximates interactions inferred using gLV when the variability in concentrations is low, matching theoretical expectations. A strong correspondence is observed between external perturbations across all datasets.

**Table 2. Number of parameters of each model.**

| Model | # interactions | # growth | # perturbations |
|---|---|---|---|
| gLV | $D \times D$ | $D$ | $D \times P$ |
| cLV | $(D-1) \times D$ | $D-1$ | $(D-1) \times P$ |
| ALR | $(D-1) \times (D-1)$ | $D-1$ | $(D-1) \times P$ |
| linear | $D \times D$ | $D$ | $D \times P$ |

Number of interaction parameters, growth parameters, and external perturbation parameters for $D$ taxa and $P$ effects.

baseline, we compared all models to a constant trajectory that predicted no change from initial conditions. A lower RMSE than the constant trajectory indicates that a model is predicting the right direction of a trajectory, as it moves away from initial conditions.

Notably, across all three datasets cLV outperformed both gLV$_{abs}$ and gLV$_{rel}$ (Fig 4). On the Diet and *C. diff* datasets, the difference can be attributed to better predictions on the first several time points as the community moves from initial conditions toward a steady state (S5 and S6 Figs). This suggests that cLV better describes relative abundances than gLV. All three models appeared to describe stable communities well. On the Antibiotic dataset, both gLV models were slower to predict recovery of the community in response to antibiotics (S7 Fig).

Our results comparing cLV to ALR and the linear model were less clear (Fig 4). On the Diet dataset, cLV better predicted community trajectories than both models ($p = 0.008$ for ALR, $p = 0.039$ for linear, one-sided Wilcoxon signed-rank test). As before, much of the difference in performance is from how well each model predicted movement from initial conditions toward a stable state (S5 Fig). All models performed similarly on the *C. diff* dataset (S6 Fig). This is likely for two reasons. First, the community converged to a stable state after few time points, which all models predicted well. Second, none of the models captured a fluctuation in community composition, where the community briefly moved away from stability, in the 5 time points immediately after introduction of *C. difficle* (S6 Fig).

On the Antibiotic dataset, we observed a slight improvement of cLV when compared to the ALR and linear models. However the result did not achieve significance. All models were slow to predict community recovery after antibiotics. However, cLV appeared to better describe the

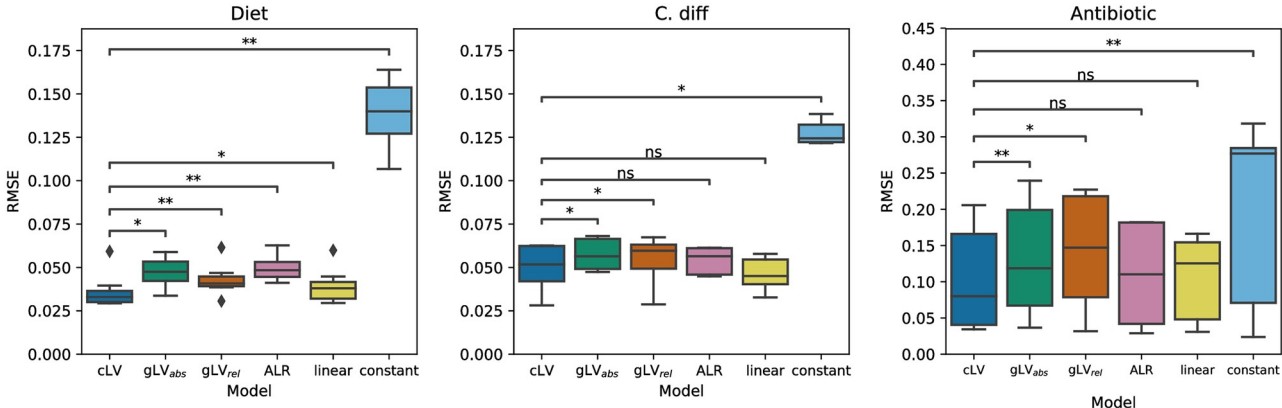

**Fig 4. Comparing predicted trajectories from initial conditions across models.** RMSE (*y*-axis) between true and estimated trajectories per sample across three datasets (panels) and six models. RMSE is computed on held out data using leave-one-out cross-validation: one sample is held out at time and the models are trained on the remaining data. Trajectories are predicted on the held out sample from initial conditions. Significance is computed relative using the one-sided Wilcoxon signed rank test (**: $p < 0.01$; *: $p < 0.05$; ns: not significant).

community after recovery than the ALR model: it more accurately predicted the final time point, empirically closest to the new stable state (S8 Fig). We did not observe a significant difference when compared to the linear model.

## Interpreting model parameters

Our derivation of cLV suggested criteria for when absolute growth rates, interactions terms, and external perturbation terms, can be recovered from relative data. Recall that the parameters for cLV are given by

$$\begin{aligned}
\bar{g}_i &\approx g_i - g_D \\
\bar{A}_{ij} &\approx A_{ij} - A_{Dj} \\
\bar{B}_{ip} &\approx B_{ip} - B_{Dp}.
\end{aligned}$$

The terms on the left are for cLV, and the terms on the right are the absolute growth rates, interactions, and external perturbations from gLV. In the section **Correspondence with the parameters of gLV** we showed that the correspondence between all but the growth rates are strong. This means we can derive criteria for when cLV will recapitulate the sign of an interaction or perturbation. We can use the former to identify cooperative or competitive interactions, and the later to identify beneficial or harmful external perturbations. For example, consider the interactions $\bar{A}_{ij} \approx A_{ij} - A_{Dj}$. We want to know when the sign of $\bar{A}_{ij}$ will be the same as the sign of $A_{ij}$. If $A_{ij}$ and $A_{Dj}$ have opposite signs, then $\bar{A}_{ij}$ will always have the same sign as $A_{ij}$. If $A_{ij}$ and $A_{Dj}$ have the same sign, then $\bar{A}_{ij}$ will have the same sign if and only if $|A_{ij}| > |A_{Dj}|$. If we assume that $A_{kl}$ is drawn from a distribution symmetric around zero, then the first and second cases are equally likely to occur—each with 0.5 probability. The first case always recapitulates the sign, while the second does so with $\Pr(|A_{ij}| > |A_{Dj}|)$. Thus, the probability cLV recapitulates the sign is greater than 0.5—better than random chance.

We can take this argument one step further, by suggesting a particular choice of denominator where $A_{Dj}$ is small. Specifically, for interactions one wants a denominator where $A_{Dj} \approx 0$, hence we want

$$\frac{d}{dt}\log \pi_i(t) - \overbrace{\frac{d}{dt}\log \pi_D(t)}^{\approx 0}$$

$$\approx \left(g_i + \sum_{j=1}^{D} A_{ij}\pi_j(t) + \sum_{p=1}^{P} B_{ip}u_p(t)\right) - \overbrace{\left(g_D + \sum_{j=1}^{D} A_{Dj}\pi_j(t) + \sum_{p=1}^{P} B_{Dp}u_p(t)\right)}^{\approx 0}$$

So we should choose a denominator where $\log \pi_D \approx const$ across all observations. For perturbations, one wants a denominator $\log \pi_D \approx const$ when the perturbation occurs. Notably, these do not need to be the same denominator: we previously showed parameters one denominator gives the parameters for all other choices.

We tested this by attempting to recapitulate an interaction network with *C. difficle* proposed by Stein *et al.* [7] on the Antibiotic dataset. By investigating learned model parameters of gLV, the authors proposed a schematic for which infection by *C. difficle* may occur. We thus wanted to see if we could suggest the same mechanism from the interaction network inferred using relative abundances and cLV. To do this, we first trained cLV using the denominator with the lowest log variance (i.e. where $\log \pi_D$ is approximately constant within a sample), and inspected interactions with respect to this denominator. We then transformed the parameters with

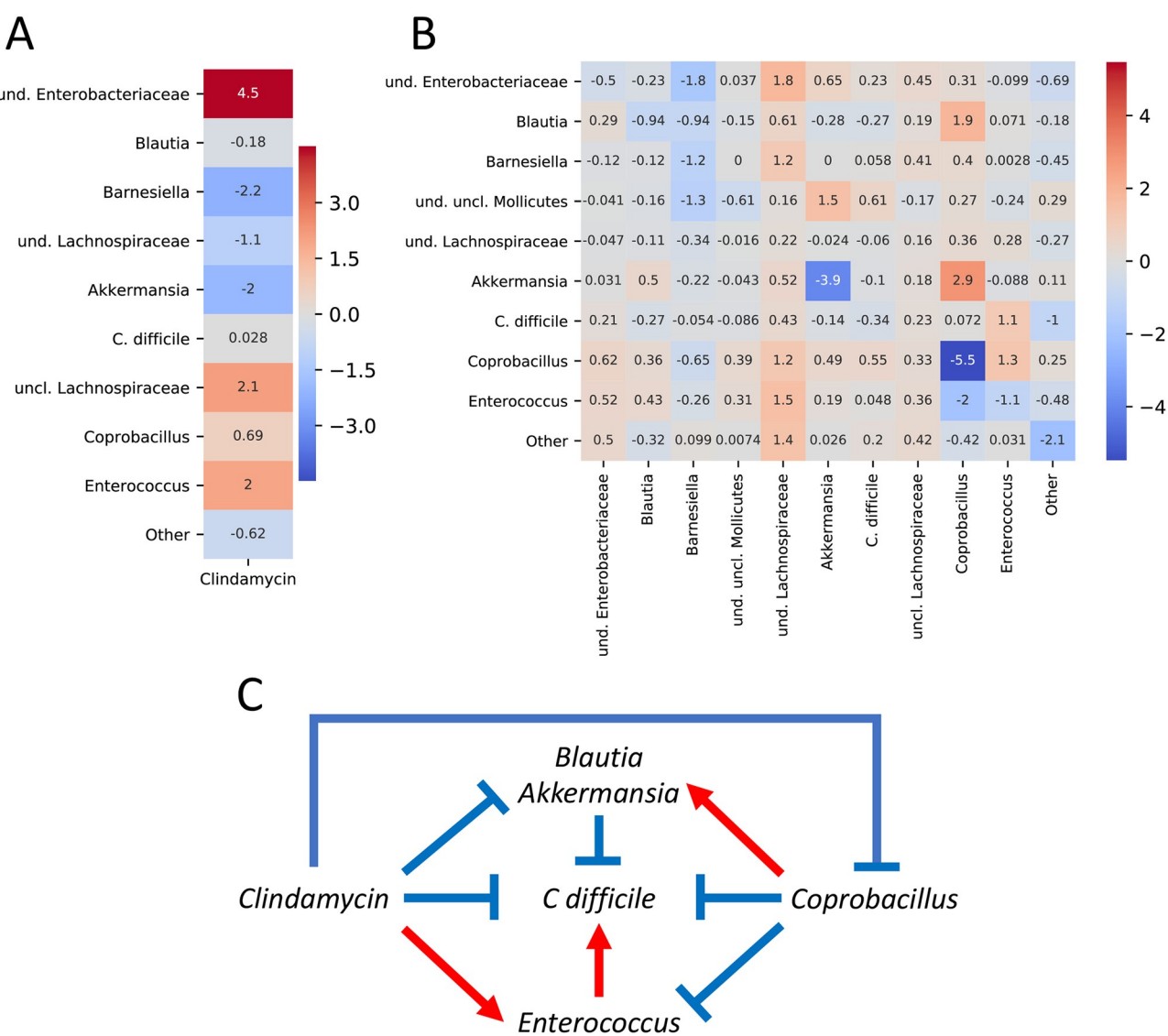

**Fig 5. cLV recapitulates absolute interactions.** Estimated effect of external perturbations (A) and interactions (B) inferred by cLV. Parameters are computed with respect to uncl. Lachnospiraceae in the denominator for interactions, and und. uncl. Mollicutes for perturbations. Estimated parameters recapitulate the interaction network with *C. difficle* (C) proposed by Stein *et al.* [7].

respect to a denominator with the lowest log variance after antibiotics, and obtained parameters for perturbations. We then inspected these parameters to see if they matched the mechanism proposed by Stein *et al.* [7].

Fig 5 displays the interaction network proposed to Stein *et al.* [7], as well as the parameters of cLV with "uncl. Lachnospiraceae" in the denominator for interactions, and "und. uncl. Mollicutes" for external perturbations. Notably, cLV recapitulates all but two of the interactions originally proposed: the effect of clindamycin on Coprobacillus and *C. difficle*. When training gLV using our inference procedure, the effect of clindamycin on Coprobacillus had the smallest magnitude of all observed effects—precisely the case where cLV will have difficulty recapitulating an effect (S9 Fig). In addition, we estimate a positive effect of clindamycin on *C. difficle* using gLV, matching the estimate by cLV. Taken together, this suggests that cLV has some utility for recapitulating absolute interactions from relative data.

## Discussion

Accurately describing microbial dynamics is crucial to understanding, modeling, and forecasting microbial communities. Here, we investigated microbial dynamics in the space of relative abundances. We introduced a new dynamical system, called compositional Lotka-Volterra (cLV), and demonstrated that cLV accurately captured relative abundance dynamics. By comparing cLV to gLV, we demonstrated a correspondence between the parameters of cLV and gLV. We leveraged this correspondence to show that cLV can sometimes recapitulate absolute interactions using relative data alone. We further evaluated how well cLV described relative abundance dynamics by comparing cLV to gLV and two other models inspired by the literature. We demonstrated that cLV more accurately predicted relative abundance trajectories than gLV, and was as good or better than the other models explored.

We derived cLV as an approximation to gLV for relative abundances, and showed that there was an approximate correspondence between the parameters of both models. The approximation depends on the variance in community size over time. Importantly, the parameters of cLV are not the same as gLV: cLV describes changes with respect to ratios between taxa, the only information provided by relative abundances. Furthermore, cLV specifically models dynamics in the constrained space of relative abundances, while gLV does not. When the variance in community size was low, we demonstrated the correspondence between the parameters of each model was strong. As the variance increased, the correspondence between interaction terms and external perturbations remained somewhat strong, but the correspondence between growth rates did not. This matched our formulation of a "signal-to-noise" ratio to measure parameter correspondence. Thus, we expect the interaction and perturbation parameters of the two models to correspond well when the variance in community size, after rescaling, is smaller than 1.

Notably, cLV more accurately forecast relative abundances than gLV across all three datasets we explored. One possible explanation is that, without a statistical model, gLV is penalized twice for sequencing noise: once for biomass estimation (e.g. qPCR) and once for relative abundance estimation (e.g. 16S sequencing). However, the effect persisted even when gLV was trained on relative abundances that eliminated one source of noise. This suggests that—if a researcher is interested in predicting relative abundances alone—no usable information is gained by access to community size data. Moreover, unless biomass is constant, our results suggest gLV is wrong model for relative abundances.

We further compared cLV to two models inspired by common assumptions in the literature: a linear model on relative abundances, and linear model under the additive log-ratio transformation. While cLV outperformed the other two models on one dataset, results were less clear on the other two. Importantly, cLV did no worse than the other models we compared, and appeared to better predict trajectories in at least some cases. Both our derivation of cLV and the results here provide some evidence that nonlinear models are required to accurately describe community dynamics. Nonetheless, our investigation is limited by lack of availability of high quality benchmark datasets. As longitudinal sampling becomes more commonplace, availability of larger high quality datasets will allow for more robust comparisons between competing methods.

Finally, we showed that in some cases cLV can recapitulate absolute interactions from relative data. We did this by recapitulating a microbe-microbe interaction network with *C. difficle* proposed by Stein *et al.* [7]. This demonstrated that similar conclusions can sometimes be drawn from relative and absolute data. While Bucci *et al.* [10] noted some ability to recapitulate the parameters of gLV when training on relative abundances, our contribution is a mathematical formulation of criteria for when absolute parameters can be recovered from relative

abundance information. Given such a formulation, it may be possible to devise a statistical test for when an absolute interaction can be recovered, and our work is a first step toward this goal.

There are several limitations to our study. In particular, our inference procedure did not incorporate technical noise due to data generation. In practice, relative abundances are estimated from sequencing counts, which can contain a considerable amount of technical noise. Indeed, our simulations showed at least some loss of accuracy in the presence of sequencing noise. While we applied a denoising step to the *C. diff* dataset, we were unable to do so on the remaining two due to differences in data collection methods and data reported. This is most likely to impact parameter estimates for rare taxa, because zero values needed to be transformed using pseudo-counts, and small differences in sequencing counts of rare taxa can cause large changes in estimated log-ratios. Nonetheless, we fit all models to data preprocessed with the same pseudo-count transformation—they all suffer from this limitation. Thus, this uniform pre-processing allowed us to compare models on equal footing.

It should also be noted that the gLV equations are not without criticism. While useful for quantifying dynamics, it is clear that they only describe an idealized system in which few real world systems abide. Additionally, influential work by Arditi and Ginzburg [31] strongly criticizes the Lotka-Volterra predator-prey model, of which gLV is a more general example. They suggest that predation (negative interactions on the A matrix) depends on ratios between taxa. Our results suggest that gLV, and cLV, described microbial trajectories well. However, an accurate model does not say anything about the physical dynamics governing the system.

Finally, a fundamental limitation of all models based on log-ratios is the inability to describe extinction and colonization. Indeed, while the compositional data analysis field has techniques for dealing with count zeros where a species falls below the detection threshold, there is no uniformly accepted technique for essential zeros (i.e. extinction; [32]). Hence, each taxon is assumed to exist at each time point. This suggests that the appropriate choice for denominator is one which does not go extinct among any time points. It may not always be possible to select a taxon uniformly present among all samples, necessitating alternate approaches to parameter inference.

Current blind spots of cLV highlight several areas for future research. One is to couple this methodology with a statistical model that includes technical and biological variation. This would allow us quantify variation not captured by the model. Another attractive extension of cLV would include extinction and recolonization, allowing more accurate forecasting of long-term trajectories where the set of taxa in a community varies. Finally, future work should focus on developing statistical methodology to recapitulate direct interactions and direct effects from relative data. By providing a theoretical understanding of microbial dynamics in the simplex, cLV represents a promising first step toward these goals.

## Methods

### Simulation evaluation

We simulated data under cLV to evaluate how well our inference procedure learned model parameters using a similar framework to Bucci *et al.* [10]. We used MDSINE [10] to obtain denoised concentrations of taxa in the *C. diff*, then rescaled estimated concentrations such that the mean community size, $N(t)$, was 1 across all samples. We then fit the parameters gLV using ridge regression with a small regularization parameter ($\lambda_A = \lambda_g = \lambda_B = 0.001$, see below). We used estimated parameters to calculate the mean and scale of growth rates ($g_{mean} > 0, \sigma_g^2$), mean and scale of self interactions ($A_{ii} < 0, \sigma_{self}^2$), the scale of between taxa interactions, $\sigma_{interact}^2$, and the mean and scale of initial concentrations ($\hat{x}_0, \sigma_{init}^2$).

For each simulation, we drew growth rates, self-interactions, and initial conditions from

$$g_i \quad \sim \mathcal{N}_+(g_{mean}, \sigma_g^2)$$
$$A_{ii} \quad \sim \mathcal{N}_-(A_{self}, \sigma_{self}^2)$$
$$x_i(0) \quad \sim \mathcal{N}(\hat{x}_0, \sigma_{init}^2)$$

We simulated sparse interactions by first drawing the probability of an interaction between taxon $i$ and taxon $j$, then drawing the interaction itself

$$z_{ij} \quad \sim \text{Bernoulli}(0.2)$$

$$A_{ij} \quad \sim \begin{cases} 0 & \text{if } z_{ij} = 0 \\ \mathcal{N}\left(0, \sigma_{interact}^2 / (\sum_j z_{ij})^2\right) & \text{if } z_{ij} = 1 \end{cases}$$

This follows estimates by Bucci *et al.* [10], who found a 20% probability of an interaction matched their real data. We further required that the resulting matrix be "stably dissapative," which guarantees existence of a steady state with all taxa present [33]. We therefore drew interaction matrix *A* repeatedly for each simulation until we found one that satisfied our criteria.

Given the parameters of gLV, we transformed them to the relative parameters of cLV using Eq 3. We then simulated noisy sequencing counts using a Dirichlet-Multinomial model with dispersion parameter 286, estimated by Bucci *et al.* [10] on the *C. diff* dataset. We evaluated model performance by computing the root-mean-square-error (RMSE) between true and estimated relative growth rates, relative interactions, and predicted trajectories from initial conditions on 5 hold out samples per simulation replicate.

We performed simulations over 30 time points, varying the sequencing depth from 1000, 2000, 5000, 10000, 25000, and noise-free. We also varied the sample size from 5, 10, 25, 50, and the time between observations from 1 day (consecutive), 2 days, 4 days, and 6 days. For each simulation replicate we required that the mean relative abundance of each taxon was greater than 0.001 and that no taxon took over the community (had abundance greater than 0.8). For each combination of parameters we performed 50 simulation replicates obtain confidence intervals and calculate significance.

## Antibiotic dataset

We downloaded and analyzed the dataset originally published by Buffie *et al.* [30] from the supplementary material of Stein *et al.* [7], who used it to investigate susceptibility to *C. difficle* infection following antibiotics. The data include bacterial concentrations from three mice populations (*n* = 9 mice, 3 from each population), along with recorded antibiotic administration, averaging 8.55 observed time points per population over 20 to 30 days. The first population served as a control and did not receive the antibiotic, the second population received a single dose of the antibiotic, and the third population received a single dose of the antibiotic followed by a *C. difficle* challenge. At each time point, a mouse from each population was euthanized, the contents of the intestine collected and, the V1-V3 16S rRNA gene was amplified and sequenced using 454 pyrosequencing. Microbial densities were calculated by quantifying 16S rRNA gene copies using quantitative PCR (qPCR). To reduce the number of missing entries (zeros), Stein *et al.* [7] modeled the top 10 most abundant genera only, and aggregated the remaining taxa into a single component marked "Other." The resulting data matrix (taxon by time-point) had fewer than 9% entries with zeros. We treated antibiotic administration as an

external perturbation, encoding the variable $u_i(t) = 1$ for $t = 1$ for populations that received the antibiotic, and $u_i(t) = 0$ elsewhere.

## Diet dataset

We downloaded and analyzed the Diet dataset from the supplementary material of Bucci *et al.* [10]. The data consist of bacterial concentrations for 13 Clostridia strains collected from fecal samples of 7 mice. Of these mice, 5 were fed a high-fiber for 2 weeks, switched to a low-fiber diet for 2 weeks, then returned to the high-fiber diet; the remaining mice were fed the high-fiber diet exclusively. The concentration for each strain was quantified separately using qPCR with taxon specific primers. Samples for diet-switched mice were collected either daily or on alternating days for 65 days, for a total of 56 observed time points. The remaining 2 mice were kept on the high-fiber diet for 5 weeks, and data collected over 29 days for a total of 25 observed time points. We treated change in diet as an external perturbation, encoding the variable $u_i(t) = 1$ during all time points when mice were switched the the low-fiber diet, with $u_i(t) = 0$ elsewhere. Fewer than 0.005% of the remaining data matrix had zeros.

## *C. diff* dataset

We downloaded and analyzed the *C. diff* dataset, which was also from Bucci *et al.* [10]. In this dataset, 5 gnotobiotic mice were orally gavaged with a bacterial mixture consisting of 22 different species. DNA sequencing data from the V4 region of the 16S rRNA gene was collected along with biomass from qPCR over the course of 56 days; there were 26 observed time points per mouse. At day 28, mice were orally gavaged with *C. difficile*. For our analysis, we used MDSINE [10] to produce denoised estimates of taxon concentrations. We set $u_i(t) = 1$ for the time point when *C. difficile* was introduced, treating it both as an external perturbation and observed taxon.

## Pseudocounts

Fitting each model requires taking a logarithm of either an observed concentration or a ratio of concentrations, with the exception of the linear model on relative abundances. We used additive smoothing (i.e. pseudocounts) on each dataset to avoid taking a logarithm of zero. To treat pseudocounts for models on relative abundances and concentrations equally, we first added pseudocounts to observed relative abundances, then transformed relative abundances to concentrations using the total concentration of a sample. Specifically, the smoothed relative abundance of a sample was

$$\hat{\pi}_i(t) = \frac{\pi_i(t) + \epsilon}{1 + \epsilon D} \tag{10}$$

The smoothed concentration was

$$\hat{x}_i(t) = N(t)\hat{\pi}_i(t) \tag{11}$$

We used pseudocounts of $\epsilon = 10^{-3}$ for each dataset. We found that models that took a log of a quantity (gLV, ALR, cLV) were sensitive to smaller pseudocounts when making predicting from initial conditions.

## Choice of denominator for cLV

We argued in the Results that the appropriate taxon for the denominator for cLV is one that is approximately log constant and common to all samples. We therefore selected a taxon for the

denominator by first computing finite-difference estimates of

$$\frac{d}{dt}\log\,\pi_i(t) \approx \frac{1}{\Delta t}(\log\,\pi_i(t) - \log\,\pi_i(t-1))$$

for all taxon $i = 1, \ldots D$ and time points $t$. We then computed the variance $\text{Var}\left(\frac{d}{dt}\log\,\pi_i(t)\right)$ for each taxon, and selected for the denominator the taxon with the lowest observed variance.

## Correspondence with the parameters of gLV

We compared parameter estimates from gLV to that of cLV on the three real datasets that included community density estimates: the Diet dataset, the *C. diff* dataset, and the Antibiotic dataset. We transformed the concentrations of each of these datasets so that the were all on the same scale, which also ensured the scale of parameters learned by cLV was approximately the same as gLV. Specifically, let $x_i(t)$ be the observed concentration of taxon $i$ and time point $t$, and let $\tilde{x}_i(t)$ be the transformed concentration. Then, if $\mu = \frac{1}{K}\sum_{k=1}^{K} x_i(t_k)$ is the mean concentration across all observed time points, the transformed concentration is

$$\tilde{x}_i(t_k) = \frac{x_i(t_k)}{\mu} \tag{12}$$

Hence $\frac{1}{K}\sum_{k=1}^{K}\tilde{x}_i(t_k) = 1$. This is equivalent to changing the units for concentration. For example, if the original units are in $10^{11}$ rRNA copies per cm$^3$ (as in [7]), then the new units are $\frac{10^{11}}{\mu}$ rRNA copies per cm$^3$. After adjusting concentrations, we then fit the parameters for both models using ridge regression. Specifically, the objective function was

$$\underset{A^*,g^*,B^*}{\arg\min}\sum_{k=1}^{K}\|g_{\text{model}}(A^*,g^*,B^*,k)\|_2^2 + \sum_{x\in\{A^*,g^*,B^*\}}\text{Penalty}(x,\alpha,\lambda_x) \tag{13}$$

with

$$\text{Penalty}(x) \quad = \lambda_x\,\|x\|_2^2 \quad x \in \{A^*,g^*,B^*\} \tag{14}$$

$$g_{\text{cLV}}(\bar{A},\bar{g},\bar{B},k) \quad = \frac{\Delta\eta_{1:D-1}(t_k)}{\Delta t_k} - (\bar{g} + \bar{A}\pi_{1:D}(t_{k-1}) + \bar{B}u(t_{k-1})) \tag{15}$$

$$g_{\text{gLV}}(A,g,B,k) = \frac{\Delta\log\,x_{1:D}(t_k)}{\Delta t_k} - (g + Ax_{1:D}(t_{k-1}) + Bu(t_{k-1})) \tag{16}$$

where $\eta_i(t) = \log\,(\pi_i(t)/\pi_D(t))$.

## Model comparison

We compared cLV to gLV and two additional models with linear interactions in additive log-ratio space and relative abundance space respectively. We fit all models using the same inference procedure, least squares with an elastic net penalty, to ensure that differences in model performance we not due to different parameter inference methods. Specifically, the objective

function is given by Eq 13, with the function $g_{\mathrm{model}}$ for the two additional models:

$$g_{\mathrm{ALR}}(\bar{A}, \bar{g}, \bar{B}, k) = \frac{\Delta \eta_{1:D-1}(t_k)}{\Delta t_k} - (\bar{g} + \bar{A}\eta_{1:D-1}(t_{k-1}) + \bar{B}u(t_{k-1})) \qquad (17)$$

$$g_{\mathrm{linear}}(A, g, B, k) = \frac{\Delta \pi_{1:D}(t_k)}{\Delta t_k} - (g + A\pi_{1:D}(t_{k-1}) + Bu(t_{k-1})) \qquad (18)$$

where $\eta_i(t) = \log(\pi_i(t)/\pi_D(t))$. The penalty term was

$$\mathrm{Penalty}(x) = \alpha\lambda_x \, \|x\|_1^2 + \alpha(1 - \lambda_x) \, \|x\|_2^2 \quad x \in \{A^*, g^*, B^*\} \qquad (19)$$

We evaluated model performance by comparing forecasted trajectories from each model starting from the same initial conditions using leave-one-out cross validation in the Antibiotic dataset, Diet dataset, and *C. diff* dataset. For each cross-validation replicate on the Antibiotic, Diet, and *C. diff* datasets, we selected regularization parameters by again performing leave-one-out cross validation on the remaining data, and selected regularization parameters that minimized the total squared error across held-out data. The regularization parameters we explore were $(\alpha, \lambda_A, \lambda_g, \lambda_B) \in Q \times R \times R \times R$ with $R = \{0.1, 0.5, 0.7, 0.9\}$ and $Q = \{0.1, 0.5, 1, 10\}$. Microbial trajectories were predicted using the RK45 numerical interaction scheme from SciPy.

## Supporting information

**S1 Appendix. Appendix to compositional Lotka-Volterra describes microbial dynamics in the simplex.**
(PDF)

**S1 Fig. Comparison between elastic net and ridge regression on simulations with sequencing noise.** Root-mean-square-error (RMSE; y-axis) between ground truth and estimated interactions, ground truth and estimated growth rates, and predicted trajectories from initial conditions on held out data across 50 simulation replicates. Community trajectories were simulated under cLV, then noisy sequencing counts to with depth of 25000 reads per sample.
(TIF)

**S2 Fig. Performance of parameter estimation with elastic net regularization with respect to sequencing depth.** Root-mean-square-error (RMSE; y-axis) between ground truth and estimated interactions, ground truth and estimated growth rates, and predicted trajectories from initial conditions on held out data across 50 simulation replicates. Community trajectories were simulated under cLV, then noisy sequencing counts with increasing sequencing depth.
(TIF)

**S3 Fig. Performance of parameter estimation with elastic net regularization with respect to time between observations.** Root-mean-square-error (RMSE; y-axis) between ground truth and estimated interactions, ground truth and estimated growth rates, and predicted trajectories from initial conditions on held out data across 50 simulation replicates. Community trajectories were simulated under cLV. Observations were selected from simulated sequenced space 1, 2, 4, or 6 days apart. Noisy sequencing counts were simulated with a depth of 25000 reads.
(TIF)

**S4 Fig. Robustness to choice of denominator.** Simulated parameters were estimated once for each taxon in the denominator per simulation replicate. The variance in RMSE (*y*-axis) across

denominators per replicate was computed to assess how choice of denominator impacted parameter estimates.
(TIF)

**S5 Fig. Ground truth and predicted trajectories on the Diet dataset.** Ground truth relative abundances (top), and predicted trajectories under each model. Each column is one sample. Scatter plots give the difference in square error per time point between each model evaluated and cLV (see *y*-label). Values above 0 (dashed line) indicate cLV is making a better prediction, while values below zero denote the opposite.
(TIF)

**S6 Fig. Ground truth and predicted trajectories on the C. diff dataset.** Ground truth relative abundances (top), and predicted trajectories under each model. Each column is one sample. Scatter plots give the difference in square error per time point between each model evaluated and cLV (see *y*-label). Values above 0 (dashed line) indicate cLV is making a better prediction, while values below zero denote the opposite.
(TIF)

**S7 Fig. Ground truth and predicted trajectories on the Antibiotic dataset.** Ground truth relative abundances (top), and predicted trajectories under each model. Each column is one sample. Scatter plots give the difference in square error per time point between each model evaluated and cLV (see *y*-label). Values above 0 (dashed line) indicate cLV is making a better prediction, while values below zero denote the opposite.
(TIF)

**S8 Fig. Model performance when predicting the final time point on the Antibiotic dataset.** RMSE (*y*-axis) between ground truth and predicted final time point for each sample across models (*x*-axis).
(TIF)

**S9 Fig. Estimated model parameters using gLV on the Antibiotic dataset.** Estimated external perturbations (A) and interactions (B) using gLV with elastic net on the Antibiotic dataset.
(TIF)

## Author Contributions

**Conceptualization:** Tyler A. Joseph, Liat Shenhav, Joao B. Xavier, Eran Halperin, Itsik Pe'er.

**Formal analysis:** Tyler A. Joseph.

**Investigation:** Tyler A. Joseph, Liat Shenhav, Itsik Pe'er.

**Methodology:** Tyler A. Joseph, Itsik Pe'er.

**Resources:** Joao B. Xavier.

**Software:** Tyler A. Joseph.

**Supervision:** Itsik Pe'er.

**Writing – original draft:** Tyler A. Joseph, Itsik Pe'er.

**Writing – review & editing:** Tyler A. Joseph, Liat Shenhav, Joao B. Xavier, Eran Halperin, Itsik Pe'er.

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
