## [Decision Letter · Decision Letter 0]

25 Nov 2019

Dear Dr Pe'er,

Thank you very much for submitting your manuscript 'Compositional Lotka-Volterra describes microbial dynamics in the simplex' for review by PLOS Computational Biology. Your manuscript has been fully evaluated by the PLOS Computational Biology editorial team and in this case also by independent peer reviewers. The reviewers appreciated the attention to an important problem, but raised some substantial concerns about the manuscript as it currently stands. While your manuscript cannot be accepted in its present form, we are willing to consider a revised version in which the issues raised by the reviewers have been adequately addressed. We cannot, of course, promise publication at that time.

Sincerely,

Vasilis Dakos, Ph. D.

Guest Editor

PLOS Computational Biology

Natalia Komarova

Deputy Editor

PLOS Computational Biology

Download Sanitized Copy

Dear  Dr Pe'er ,

As you will see  the two expert reviewers that evaluated your manuscript, they both find the manuscript of interest, but they raised some considerations in regards to its conclusions and analysis. In your revision, you should pay attention to some of their statements at the discussion as highlighted by Rev2, and in particular to provide a better justification of the relevance of your work for clinicians as suggested by Rev1. In this way, the manuscript can become much more than an improved computational contribution. In addition, I share the concerns raised on the usefulness of section 2.3 and 2.6 and the specifics of the model fitting raised by both reviewers, as well the importance of noise and choice of denominator data. The two reviewers have also raised a series of other comments that need to be addressed to make the paper stronger and more impactful.

Reviewer's Responses to Questions

**Comments to the Authors:**

Reviewer #1: In this manuscript, Tyler A. Joseph and colleagues adopt ecology models of community dynamics to a compositional space. The paper is clearly written, and the topic will likely be of at least modest interest to the community of researchers interested in the theory of how to apply ecological principles to sequence count data. It is less clear that the progress in theory the paper represents will be of immediate use to the larger community of researchers who have longitudinal 16S or WGS-metagenome data and are searching for ways to relate these data to clinical parameters.

For table #1, could the authors add columns indicating the total sample size for each study, the # of free parameters in each corresponding model and the method used for generating absolute abundance in each study. In general, the authors could be more explicit about the relationship between the number of free parameters in the models they used and the sample size. It is unclear from the current version of the manuscript if the models tend to have nearly as many or more parameters than the number of samples. In such cases, good fits to the data are not hard to achieve (as the physicist von Neumann was alleged to have said, “With four parameters I can fit an elephant, and with five I can make him wiggle his trunk.”). Some consideration of model flexibility given model complexity should be added to the discussion. Do the cLV and gLV and linear models (equations (11) and (12)) all have the same number of free parameters? Or can the modest increases in performance seen by cLV in fitting the data be explained by an increase in the number of free parameters?

The glv equations are not universally appreciated as being appropriate for ecological modeling and some discussion of this should be added to the introduction or discussion (for example J. theor. Biol. (1989) 139, 311-326; Coupling in Predator-Prey Dynamics: Ratio-Dependence; ROGER ARDITIt AND LEV R. G1NZBURG).

The section on “clinical relevance” is not particularly compelling. It’s not immediately clear what an ROC based on “the greatest predicted fraction of Enterococcus (across all predictions) as the prediction threshold” could mean. This seems a bit circular as the model is apparently being used to measure its own predictions. This section (section 2.6) could potentially be removed in its entirety. Alternatively, the authors could find a more traditional clinical prediction (case/control status, probability of getting an infection) or some other feature more clearly related to a patient health outcome as a test of their model. They could also more directly demonstrate the utility of their model by directly comparing to more frequently used data analysis schemes (for example, how does their model do when compared to a random forest classifier built with leave X out classification).

Could the authors expand their discussion of the choice of the denominator taxa in their method? From the methods section, different strategies were taken for choosing the denominator in the different datasets. Could the authors demonstrate that the superiority of the cLV method was robust to these choices. That is to say, if a different choice for the denominator was used, does cLV still out-perform the other methods?

Reviewer #2: attachment

**Have all data underlying the figures and results presented in the manuscript been provided?**

Reviewer #1: Yes

Reviewer #2: Yes

PLOS authors have the option to publish the peer review history of their article (what does this mean?). If published, this will include your full peer review and any attached files.

Reviewer #1: No

Reviewer #2: Yes: Georg K. Gerber

---

## [Decision Letter · Decision Letter 1]

17 Mar 2020

Dear Dr Pe'er ,

Thank you very much for submitting your manuscript "Compositional Lotka-Volterra describes microbial dynamics in the simplex" for consideration at PLOS Computational Biology. As with all papers reviewed by the journal, your manuscript was reviewed by members of the editorial board and by several independent reviewers. The reviewers appreciated the attention to an important topic. Based on the reviews, we are likely to accept this manuscript for publication, providing that you modify the manuscript according to the review recommendations.

The two expert reviewers have both found that your revision adequately addressed their raised concerns.

I would suggest that you make one more effort to address the few issues raised by Rev 2 that I think would increase the clarity of your work and could potentially increase the impact of your contribution.

Sincerely,

Vasilis Dakos, Ph. D.

Guest Editor

PLOS Computational Biology

Natalia Komarova

Deputy Editor

PLOS Computational Biology

[LINK]

Dear Dr Pe'er ,

The two expert reviewers have both found that your revision adequately addressed their raised concerns.

I would suggest that you make one more effort to address the few issues raised by Rev 2 that I think would increase the clarity of your work and could potentially increase the impact of your contribution.

Best regards,

Vasilis Dakos

Reviewer's Responses to Questions

**Comments to the Authors:**

Reviewer #1: The authors have done a good job of responding to the technical queries of the reviewers. The paper remains a solid - if somewhat esoteric - contribution to the literature and I have no further suggestions for improvements.

Reviewer #2: The authors have done substantial work to improve the manuscript and have addressed all of my major concerns. In particular, I really appreciate the authors’ work deriving new qualitative and quantitative understanding of what can be inferred by their model (e.g., the new Section 2.6). The microbiome field really needs this type of careful and thoughtful analysis to get beyond blind application of generic statistical and machine learning methods.

At this point, my critique involves changes for clarity and consistency in the manuscript.

Also, I’d like to apologize for the amount of time it took me to re-review. I received the manuscript while on a tight grant deadline and informed the editorial office I wouldn’t be able to re-review until the first week in March. However, due to the COVID-19 situation, both personally and events at my institution, I’ve been delayed in getting this done.

Comments:

Pg 2, starting line 35: “We show that relative abundances are sufficient to learn the process governing microbial dynamics…”

I would cut this sentence or revise. As written, this sentence could be misinterpreted as: 1) you’re learning the actual physical process (rather than a model) and, 2) absolute abundance measurements provide no useful information.

Pg 4, line 79: “Yet, binning taxa into quantiles loses fine-grained information…”

I would soften this to “MAY lose fine-grained information…”

This depends on the discretization scheme and what the model is trying to learn. For relationships such as signs of interactions, discretization approaches could potentially work quite well.

Pg 5, line 107: “…recapitulate a mechanism of C. difficle [typo] colonization…”

I wouldn’t really say the cited work offers a mechanism of C. difficile colonization. Maybe clearer to call it a “proposed directed microbe-microbe interaction network with C. difficile” or something similar, since that’s what you’re ultimately comparing to. This same phrase is used elsewhere in the manuscript, so should be changed throughout.

Pg 8, line 166: This is a really nice insight! It allows one to easily see what the “compositional” contribution is in these types of models.

Pg 9, line 186: “…form of equation (6) makes direct application of these methods challenging…”

I would revise to “…form of our model (e.g., equation 6) does not allow us to readily apply these methods…”

The model in Bucci et al is fully Bayesian and hierarchical. It’s not clear to me how the cLV model could best be recast as a fully Bayesian model (or that equation 6 is the only issue) and what inference method would be most efficient. That’s an interesting question for future research!

Pg 9, line 207: “…we wanted to ensure that difference [typo] choices of denominator do not affect quality of inference.”

This is a very nice result that demonstrates robustness of the method.

Pg 10, line 228: “…“Diet” dataset included 7 mice. 5 mice were fed from a high-fiber diet…”

Just to clarify, this dataset only contains absolute abundances. The concentrations of the individual taxa were measured via qPCR. Relative abundances weren’t measured in these experiments.

Pg 10, line 238: “For this particular task, we chose to use ridge regression since elastic net may choose to zero out different parameters for each model, making direct comparison challenging.”

I’m not sure what’s meant here or why ridge regression was used here and elastic net was used elsewhere. Both ridge regression and elastic net are shrinkage estimators that don’t zero out regression coefficients (whereas the lasso algorithm does), but just bias them to small values. So, why won’t elastic net work just as well here? Based on your analyses on simulated data, it doesn’t seem there’s a meaningful difference in performance between the two methods. So, couldn’t you just use ridge regression throughout the manuscript for consistency? Or, are you using some version of elastic net that actually does zero out parameters?

Pg 11, line 248: “Nonetheless, correspondence between interactions and external effects was strong among the three datasets we explored.”

This is another nice result that empirically confirms when compositional effects may be important and when they’re not.

Pg 12, line 262: “To avoid overfitting, which would cause models with more parameters to perform better, we only evaluated predicted trajectories on held out test data using leave-one-out cross validation. That is, for each dataset we held out one sample at time, and trained models on the remaining data.”

As written, I think this is a bit confusing as to the reason why you’re doing cross-validation. Maybe something like: “We evaluated models based on a measure of generalization performance, or the ability to predict unseen data. Generalization performance metrics inherently penalizes models that overfit, or use parameters to fit noise in data rather than model actual signal. These metrics allow for principled comparison of models with different structures or numbers of parameters. In our case, we used a metric that evaluated predicted trajectories on held out test data via leave-one-out cross validation. That is, for each dataset we held out one time-series in turn, and trained models on the remaining data.”

Pg 13, line 284: “Second, none of the models captured a community disturbance resulting from the introduction of C. difficle [typo].”

I continue to find this statement confusing. What is meant by the models not capturing a community disturbance? Since you’re modeling C. diff as one of the microbes in the ecosystem, doesn’t that capture the disturbance? I think more clearly defining what’s meant by a community disturbance (in terms of your model and this particular dataset) is needed.

Pg 13, line 292: “…suggested criteria for when an absolute term…”

I’d explicitly state “absolute growth rates and interaction terms” since it’s unclear what “term” means until later in the paragraph.

Pg 13, line 298: I don’t entirely follow the logic that the probability of getting the sign correct is 75%. It’s not clear to me that the assumption of a symmetric distribution around zero implies equal probability of the four cases. Doesn’t it depend on P(|A_{ij}| > |A_{D_{j}}|)?

Pg 14, line 305: This paragraph is overall a bit confusing. Perhaps the term “optimal” rather than “right” would be better. Also, the statement that “…these do not need to be the same denominator…” is confusing. I think some copyediting will help here.

Pg 15, line 347: “Notably, cLV more accurately forecast community trajectories than gLV across all three datasets we explored.”

Be clear that the forecasting task is relative abundances, i.e., “Notably, cLV more accurately forecast community trajectories of relative abundances than gLV across all three datasets we explored.”

Pg 15, line 351: “One explanation for the discrepancy is that gLV is penalized twice for noisy data, while cLV only once.”

I disagree that this is an inherent problem with the gLV equations. It relates more to the statistical noise model (or lack thereof.) You’re comparing two models that lack explicit noise models. Models using either gLV or cLV dynamics could use explicit noise models. This has already been shown to benefit gLV-based models. It would likely benefit a cLV model as well. The issue as to whether absolute abundances can be accurately measured is a separate point. There are technical challenges to making these measurements and the first methods employed were fairly poor. But, there have already been and continue to be improvements in the experimental methodologies.

I think a more interesting and relevant question from the computational perspective (and for purposes of this manuscript) is why gLV underperforms when it’s given only relative abundance information. As you’ve pointed out, this makes an assumption of constant biomass, which is clearly wrong in many cases. Again, as you’ve pointed out, gLV isn’t a model for compositional data. So, unless there’s constant biomass, it’s the wrong model for relative abundances. This is the most important take-away for readers. The intricacies of experimental technologies and statistical error models are secondary issues.

Pg 16, Line 372: “…our contribution is a formulation of when parameters can be recovered mathematically…”

This is a bit hard to follow. Maybe something like “..our contribution is a mathematical formulation of criteria for when absolute parameters can be recovered from relative abundance information.”

Regarding the C. diff dataset in Bucci et al:

Just to be clear, because this was a gnotobiotic experiment, we knew exactly which taxa were introduced, and from the sequencing data, we did not detect any contaminants. Any additional OTUs beyond the taxa actually present are bioinformatic artifacts and occur at very low abundances. So, in your analyses, it makes most sense to include all of the taxa experimentally introduced and exclude any bioinformatic artifacts, as done in the original MDSINE analyses. The MDSINE analyses were done using older bioinformatics pipelines. In recent gnotobiotic experiments we’ve done with the same taxa, we can recover nearly exactly the same ASVs/OTUs as there are actual taxa.

General: there are a number of typos throughout, particularly in the new sections. Careful copyediting is needed.

**Have all data underlying the figures and results presented in the manuscript been provided?**

Reviewer #1: Yes

Reviewer #2: Yes

PLOS authors have the option to publish the peer review history of their article (what does this mean?). If published, this will include your full peer review and any attached files.

Reviewer #1: No

Reviewer #2: Yes: Georg K. Gerber
---

## [Editor Report · Decision Letter 2]

28 Apr 2020

Dear Dr Pe'er,

We are pleased to inform you that your manuscript 'Compositional Lotka-Volterra describes microbial dynamics in the simplex' has been provisionally accepted for publication in PLOS Computational Biology.

Best regards,

Vasilis Dakos, Ph. D.

Guest Editor

PLOS Computational Biology

Natalia Komarova

Deputy Editor

PLOS Computational Biology

Editors remarks:

I find that your responses to the Reviewers comments were satisfactory and your manuscript has benefited.

I would only like to point your attention in taking the time to reread your text to correct minor mistakes - eg lines 82 to explore, 90 - ignores, 427 strange syntax- are some examples that I came across while reading through.

---

## [Editor Report · Acceptance letter]

22 May 2020

PCOMPBIOL-D-19-01680R2 

Compositional Lotka-Volterra describes microbial dynamics in the simplex

Dear Dr Pe'er,

I am pleased to inform you that your manuscript has been formally accepted for publication in PLOS Computational Biology. Your manuscript is now with our production department and you will be notified of the publication date in due course.

With kind regards,

Sarah Hammond
